# Optimization and Validation of an Adjustable Activity Classification Algorithm for Assessment of Physical Behavior in Elderly

**DOI:** 10.3390/s19245344

**Published:** 2019-12-04

**Authors:** Wouter Bijnens, Jos Aarts, An Stevens, Darcy Ummels, Kenneth Meijer

**Affiliations:** 1Instrument Development, Engineering and Evaluation, Maastricht University, P.O. Box 616, 6200 MD Maastricht, The Netherlands; 2Research Centre for Autonomy and Participation of Persons with a Chronic Illness, Zuyd University of Applied Sciences, P.O. Box 550, 6419 DJ Heerlen, The Netherlands; 3Department of Nutrition and Movement Sciences, Maastricht University, P.O. Box 616, 6200 MD Maastricht, The Netherlands

**Keywords:** accelerometers, algorithm, validation, elderly, physical activity, sedentary behavior

## Abstract

Due to a lack of transparency in both algorithm and validation methodology, it is difficult for researchers and clinicians to select the appropriate tracker for their application. The aim of this work is to transparently present an adjustable physical activity classification algorithm that discriminates between dynamic, standing, and sedentary behavior. By means of easily adjustable parameters, the algorithm performance can be optimized for applications using different target populations and locations for tracker wear. Concerning an elderly target population with a tracker worn on the upper leg, the algorithm is optimized and validated under simulated free-living conditions. The fixed activity protocol (FAP) is performed by 20 participants; the simulated free-living protocol (SFP) involves another 20. Data segmentation window size and amount of physical activity threshold are optimized. The sensor orientation threshold does not vary. The validation of the algorithm is performed on 10 participants who perform the FAP and on 10 participants who perform the SFP. Percentage error (PE) and absolute percentage error (APE) are used to assess the algorithm performance. Standing and sedentary behavior are classified within acceptable limits (±10% error) both under fixed and simulated free-living conditions. Dynamic behavior is within acceptable limits under fixed conditions but has some limitations under simulated free-living conditions. We propose that this approach should be adopted by developers of activity trackers to facilitate the activity tracker selection process for researchers and clinicians. Furthermore, we are convinced that the adjustable algorithm potentially could contribute to the fast realization of new applications.

## 1. Introduction

Digital innovations can promote and support an increase in physical activity and a reduction of sedentary time and thereby improve the health, well-being and participation of all individuals, according to the WHO [1]. Physical activity trackers are such digital innovations and have been successfully used in different applications to classify physical activity in terms of three key outcome measures: dynamic, standing and sedentary behavior [2,3,4]. Actually, the number of possible applications is enormous, taking in to account the various target populations, tracker wear locations and outcome measures [5,6,7,8].

To select or improve a physical activity tracker for these new applications, it is important to understand their strengths and limitations. Activity trackers do not measure physical activity directly. Using sensor technology, e.g., accelerometers, they measure the movement of the body (segment). Then, they estimate physical activity by applying application-specific algorithms to this raw data [9]. Ideally, both sensor and algorithm specifications are well described by the manufacturer or are available in literature. However, due to the abundance of methods, protocols and measures of validity, combined with a lack of transparency on the algorithm methodology, it is difficult for researchers and clinicians to compare different physical activity trackers [9,10]. To overcome this, the validation procedure should be described transparently, include at least a simulated free-living protocol (which corresponds to the main daily activities), and also should be performed by the target population [9,11]. Furthermore, the underlying physical activity algorithms should be presented with sufficient detail, including methodological choices and their implications. Unfortunately many classification algorithms remain a black box.

Technically, when activity trackers are not transparently described, every new physical activity classification application would require validation of the activity tracker’s algorithm performance and, when not accurate enough, development of a new algorithm [12,13,14,15]. This is a time- and resource-consuming process. Nevertheless, most available algorithms are built with the same conceptual building blocks, i.e., (1) a preprocessing phase to remove artifacts from the raw data, (2) data segmentation, (3) extraction of data features, and (4) a classifier which are used to translate the raw data into interpretable outcome measures [16,17,18,19,20,21]. Additionally, according to Bersch et al., the sampling rate, data segmentation method, window size and classifier method are aspects that affect algorithm accuracy the most [17]. Using these building blocks, Annegarn et al. presented and validated an activity classification algorithm for chronic organ failure patients [22]. We are convinced that through the easily adjustable algorithm parameters (i.e., data segmentation window size and data feature threshold values used by the classifier), it is possible to optimize the performance of this algorithm for different target populations and the locations where the tracker is worn on the body.

Our aim is to transparently present an adjustable physical activity classification algorithm which discriminates between dynamic, standing, and sedentary behavior. We demonstrate how the parameter settings are optimized for a healthy elderly target population and tracker worn on an upper leg location. Finally, we validate the algorithm in a simulated free-living environment and compare it to the algorithm settings used by Annegarn et al. This method could contribute to a well-founded physical activity tracker selection process as well as to a fast realization of other new applications.

## 2. Materials and Methods

### 2.1. Study Design

A cross validation study was performed on forty healthy elderly. Twenty participants performed a fixed activity protocol (FAP), the other twenty participants performed a simulated free-living protocol (SFP). The optimization of the parameter settings of the adjustable algorithm was performed on a random selection of twenty participants (ten performing the FAP and ten performing the SFP protocol). The validation of the optimized algorithm was performed in the remaining twenty participants. The study design is presented in Figure 1.

This study was approved by the local ethics board Atrium–Orbis–Zuyd Medical Ethical Committee (METCZ20180012). All participants provided written informed consent.

### 2.2. Participants

Healthy elderly participants were recruited from several local elderly associations. Participants were eligible for inclusion if they were community dwelling, at least 65 years old, and did not meet the Dutch physical activity guidelines [23]. Insufficient understanding of the Dutch language, use of a walking aid, and an asymmetrical gait were the exclusion criteria.

### 2.3. Device Description

The MOX Activity Logger (MOX; Maastricht Instruments, Maastricht, The Netherlands) is the successor of the DAAFB [5,24] and CAM [7,22,25] activity loggers. The device contains a tri-axial accelerometer sensor (ADXL362; Analog Devices, Norwood, MA, USA) in a small waterproof housing (35 × 35 × 10 mm, 11 g) [26]. The MOX uses a custom-made, double-sided, waterproof patch for body attachment. Raw acceleration data (±8 g) is measured in three orthogonal sensor axes (X, Y and Z) at a 25 Hz sampling rate and stored directly on the internal memory. The MOX is capable of measuring and storing data continuously up to seven days. Data analysis is performed offline. The MOX has been successfully used for physical activity monitoring in colorectal cancer survivors and COPD patients [3,27] worn on an upper leg location.

### 2.4. Data Collection

The data was collected in the Human Performance Laboratory of Maastricht University (Maastricht, The Netherlands) or at Zuyd University of Applied Science (Heerlen, The Netherlands). After providing informed consent, demographic data were collected (gender, age, body weight, body length). Next, the MOX was attached on the upper part of the non-dominant leg about 15 cm above the knee, as shown in Figure A1 (Appendix B). Participants were video recorded during the execution of the protocol. These recordings served as the gold standard for the classification of physical activities (dynamic, standing or sedentary behavior) [11]. After the execution of the protocol, the data was downloaded from the MOX and stored on a secured internal data server, together with the video recordings. A 10 m walk test was included in both protocols as a standardized test to determine the average comfortable walking speed [28].

### 2.5. Activity Protocol

Two different protocols were developed: a fixed activity protocol (FAP) and a simulated free-living protocol (SFP). The FAP consisted of a predefined sequence (order and duration) of different types of activities as shown in Figure 2a.

The SFP protocol consisted of a set of different target population specific activities of daily living (ADL). Participants were free to choose the order and duration for each type of activity. Figure 2b shows the activities and an example of their order and duration. The location of the activities was spread throughout a 120 m^2^ laboratory, resulting in different durations of the transfers between different activities. This protocol was used to simulate a real free-living situation. Both at the beginning and the end of each protocol, a squat movement was performed for post-hoc synchronization between the video recording and the acceleration data.

### 2.6. Algorithm Description

The physical activity classification algorithm originates from the algorithm developed by Annegarn et al., used for chronic organ failure patients and worn on an upper leg location [22]. The adjustable physical activity classification algorithm, starting from the raw tri-axial acceleration data, uses the same decision tree classifier. Additionally, three parameters have been made to be easily adjustable to optimize the algorithm for specific applications in terms of target population and wear location. These parameters are data segmentation window size, amount of physical activity threshold, and sensor orientation threshold. The algorithm for the tracker worn on an upper leg location discriminates between (1) dynamic, (2) standing, and (3) sedentary (lying/sitting) behavior. A schematic overview of the algorithm is shown in Figure 3.

The algorithm’s decision tree starts with the raw tri-axial acceleration data that, for each sensor axis (X, Y and Z), contains three components: (1) noise acceleration (NA), (2) body acceleration (BA), and (3) gravitational acceleration (GA). As NA originates from sensor noise and not from physical activity it is eliminated in the preprocessing filtering step by applying a moving average filter with a filtering window size of 0.12 s to the raw acceleration data. Next, the acceleration data is segmented into windows with a fixed non-overlapping sliding window method [17]. The data segmentation window size (WS) is the first adjustable parameter.

Next, the amount of physical activity is determined for each window. This is defined as the signal magnitude area (SMA) of the BA and expressed in counts per second (cps) [18,19,20,21]. Regarding this, the GA is eliminated from the acceleration data by applying a fourth order Butterworth High Pass Filter with a cut-off frequency of 1 Hz. Then, the absolute value of the remaining BA is summed over all three axes and over all data samples that make up a complete window. This calculation is shown in Formula (1):
(1)$$SMA_{j}=\overset{j*N}{\underset{i=\left(\left(j-1\right)*N+1\right)}{\sum }}{\left|BA\right|}_{i}$$
where N is the data segmentation window size times the sampling rate, j is the index for each window, and i is the index for each data sample. Based on the threshold value for this amount of physical activity (PA Th), which is the second adjustable parameter, each window is classified as dynamic or static.

Concerning the static windows, the sensor orientation is determined. This is defined by the mean GA for the three axes individually. Considering this, the BA is eliminated from the acceleration data by applying a fourth order Butterworth Low Pass Filter with a cut off frequency of 1.25 Hz. Then, for each axis the remaining GA is averaged over all data samples that make up a complete window. Regarding the X-axis, this calculation is shown in Formula (2):
(2)$$GAx_{j}=1/N\overset{j*N}{\underset{i=\left(\left(j-1\right)*N+1\right)}{\sum }}GAx_{i}$$
where N, i and j have the same definition as in Formula (1). Given a certain wear location of the sensor, based on a threshold value for the sensor orientation (SO Th), which is the third adjustable parameter, the body posture can be classified for each static window. Concerning the tracker worn on an upper leg location described in the current work, the threshold value for sensor orientation in the X-axis discriminates between the body postures: standing and sedentary (sitting/lying) [29]. The remainder of this article has the sensor orientation in the X-axis referred to as sensor orientation.

### 2.7. Data Analysis

Data analysis of participant characteristics was performed using Prism (GraphPad Prism 8.2.1(441); GraphPad Software, San Diego, CA, USA). Descriptive statistics of the participant characteristics were presented as the number (percentage) for the categorical variable gender and as a median (95% CI) for the continuous variables age, body length, weight, and average walking speed. Due to the limited amount of participants in each sub-group (n = 10, see Figure 1) non-parametric Mann–Whitney U tests were applied to assess differences between the four sub-groups of the study design.

All video recordings of the simulated free-living protocol (SFP) were classified as dynamic, standing or sedentary behavior by two independent observers using the Dartfish EasyTag-Note, creating an event-based spreadsheet where each second was classified [30]. The definition used to classify each activity is shown in Figure 2. Both observers were blinded to the classifications made by the other observer and by the algorithm. The inter-observer reliability was assessed with the intraclass correlation coefficient (two-way random, absolute agreement) and Bland–Altman plots.

The raw acceleration data was analyzed using MATLAB (R2018b; The MathWorks Inc., Natick, MA, USA). To obtain a fixed protocol (FAP) measurement, the acceleration data was cut into pieces based on the video recording, resulting in only one type of activity per data part. To obtain a simulated free-living protocol (SFL) measurement, all acceleration data was analyzed as one complete data part. Regarding these data parts, the adjustable physical activity classification algorithm was used to calculate dynamic, standing and sedentary time.

#### 2.7.1. Parameter Setting Optimization

The data of the optimization group was used to calculate dynamic, standing and sedentary time for a variety of settings of the adjustable parameters. This was done to assess the performance of the algorithm in terms of its optimal classification error. Optimization started with the fixed parameter settings (WS = 1 s, PA Th = 5 cps, SO Th = 0.8 g) incorporated in the algorithm developed by Annegarn et al. Concerning the described wear location (upper leg), a sensor orientation threshold of 0.8 g discriminated standing from sedentary behavior. This corresponded to an angle of 36° between the gravity vector and the longitudinal direction of the upper leg. Since there was no reason to assume that there was a better setting for the healthy, elderly population that wore the tracker in the same location, this parameter was retained in the current work.

Regarding the data segmentation window size, parameter settings varying from 0.25 s up to 6.7 s and even 74 s have been described [17,21]. Window sizes from 1 s up to 10 s were evaluated, using a PA Th of 5 cps. These settings were chosen based on the type and duration of activities carried out by the target population. Actually, the healthy elderly tended to be sedentary with short periods of physical activity. Then, for the optimal data segmentation window size setting, the threshold for the amount of physical activity was evaluated from 3 cps up to 12 cps to determine the optimal amount of physical activity threshold and complete the set of optimal parameter settings.

Classification error of the algorithm was assessed using percentage error (PE) and absolute percentage error (APE):(3)$$PE=\frac{Tot\;Time\;Activity\;Class_{Video_{}\;}-\;Tot\;Time\;Activity\;Class_{MOX_{}}}{Tot\;Time\;Activity\;Class_{Video_{}}}*100$$
(4)$$APE=\;\frac{\left|Tot\;Time\;Activity\;Class_{Video_{}\;}-\;Tot\;Time\;Activity\;Class_{Mox_{}}\right|}{Tot\;Time\;Activity\;Class_{Video_{}}}*100$$
where, $Tot\;Time\;Activity\;Class_{Video_{}\;}$ is assessed by Observer 1. PE reflects the error between the video recordings and the MOX classification algorithm on a group level. Since APE does not cancel out errors from over- and underestimation, it reflects errors on an individual level. As PE and APE are relative measures, it is possible to compare them across different studies [9]. PE and APE values are expressed as median and 95% confidence interval (CI).

#### 2.7.2. Algorithm Validation

After the parameter settings were optimized, the data of the remaining twenty participants was used for validation. The performance of the algorithm was assessed for the fixed and simulated free-living protocol separately and, subsequently, classified with the algorithm settings used by Annegarn et al [22].

Agreement values between 90–110% were considered acceptable [31]. Additionally, Bland–Altman plots with limits agreement were used to assess potential bias between the video recordings and the algorithm classification.

## 3. Results

### 3.1. Participant Characteristics

Forty healthy elderly were recruited for this study. A random selection of twenty participants was selected for the optimization of the algorithm. The remaining twenty participants were used for the validation of the algorithm. The participant characteristics are shown in Table 1.

According to the Mann–Whitney U test, participant characteristics did not significantly differ.

### 3.2. Inter-Observer Reliability Simulated Free-Living Protocol

The inter-observer reliability of the video recording used to assess dynamic, standing, and sedentary behavior was high: 0.96, 0.99 and, 1.0 respectively (ICC Agreement).

The limits of agreement for dynamic behavior (−48.9 s to 40.5 s), standing (−48.9 s to 39.8 s), and sedentary behavior (−26.4 s to 28.1 s) were narrow and showed no systematic differences. The Bland–Altman plots are shown in Figure A2.

### 3.3. Parameter Setting Optimization

#### 3.3.1. Data Segmentation Window Size

To find the optimal data segmentation window size, ten different windows ranging from 1 s up to 10 s were tested using an amount of physical activity threshold of 5 cps. The results for all data parts of the optimization data set are shown in Figure 4a,b. The median and 95% CI of the PE and APE values for the three best performing window sizes, 1–3 s, can be found in Table 2.

A window size of 2 s showed the smallest classification error for classifying dynamic (PE: 0.6% (0.4–0.7), APE: 0.9% (0.6–1.0)), standing (PE: 6.8% (4.4–13.4), APE: 6.8% (5.1–13.4)), and sedentary (PE: 2.9% (1.8–4.2), APE: 2.9% (1.8–4.2)) behavior. For more detailed information see Appendix A.

#### 3.3.2. Amount of Physical Activity Threshold

To optimize the amount of physical activity threshold, values from 3 cps to 12 cps were tested using a window size of 2 s. The results for each threshold are shown in Figure 4c,d. PE and APE values (median and 95% CI) for the three best performing thresholds can be found in Table 2.

A threshold of 7 cps further improved the APE classification error for standing and sedentary behavior (3.9% (2.2–6.4) and 2.0% (1.4–2.5), respectively). Concerning dynamic behavior, the error did not change (0.9% (0.7–1.0)). For more detailed information see Appendix A.

### 3.4. Algorithm Validation

To validate the algorithm, the parameter settings of the optimized algorithm were: (1) data segmentation window size of 2 s, (2) amount of physical activity threshold of 7 cps and (3) sensor orientation threshold of 0.8 g.

The PE and APE results for all data parts of the validation data set are shown in Figure 5 for the fixed activity (FAP) and the simulated free-living (SFP) protocols separately. The upper part of Table 3 shows that the median of all PE and APE values for all classifications are within the proposed limits, except for the dynamic classification for the SFP protocol—this is slightly higher than acceptable. Regarding the SFP, the confidence intervals are wider compared to the FAP.

Within the FAP, the four dynamic activities were separately classified with acceptable errors: 2 km/h walking (0.6% (0.4–0.9)), 4 km/h (0.6% (0.1–0.8)), overground walking (0.6% (0.3–1.1)), and biking (0.7% (0.3–1.0)). The same holds for the two sedentary activities: sitting (2.9% (1.2–5.9)) and lying (2.2% (0.9–3.1)). The Bland–Altman plots with limits of agreement can be found in Figure A3. For more detailed information see Appendix A.

The PE and APE values for the classification with the algorithm settings used by Annegarn et al. (WS = 1 s, PA Th = 5 cps, SO Th = 0.8 g) are shown in the lower part of Table 3 [22]. The classification error is higher than the preset limit for the SFP of both dynamic and standing behavior. Also, the 2 km/h walking (12.9% (1.0–34.1)) from the FAP was classified with a higher error than accepted. All other activities included in FAP showed acceptable errors (4 km/h walking (0.5% (0.1–1.2)), overground walking (2.6% (1.1–6.0)), biking (0.5% (0.2–2.0)), sitting (1.9% (1.1–3.4)), lying (1.5% (0.8–2.0)), and standing (1.6% (1.5–2.4))).

## 4. Discussion

The aim of the current work was to transparently present an adjustable physical activity classification algorithm which discriminates between dynamic, standing, and sedentary behavior. The algorithm provides easily adjustable parameters (data segmentation window size, amount of physical activity threshold, and sensor orientation threshold) to optimize the classification performance for applications using different target populations and wear locations. It was found that, for a healthy elderly population and a tracker worn on an upper leg location, validation of the optimized parameter settings showed good results for the fixed activity protocol (FAP) and the simulated free-living protocol (SFP). Furthermore, it was shown that parameter settings that were validated for patients with chronic organ failure by Annegarn et al. did not yield an optimal parameter set for healthy community dwelling elderly [22].

Banos et al. showed that optimizing the data segmentation window size will minimize the chances of combining different activities of daily living (ADLs) within one window or split into different windows [16]. Regarding the application under investigation, a data segmentation window size of 10 s leads to classification errors of up to 80%. Compared to previous studies, a data segmentation window size of 2 s showed the smallest error with a narrow 95% CI [16,21,32]. The smallest tested data segmentation window size (1 s, used by Annegarn et al. [22]) showed a wider 95% CI compared to the optimal window size. An adjustable amount of physical activity threshold is needed since different target populations will perform the same kind of activity with different intensities and, thus, accumulate different amounts of physical activity. Considering the overground walking, for example, the speed and cadence differed significantly between 30- and 90-year-olds [33]. The current work showed that for healthy elderly, a PA Th of 7 cps provides better results than a PA Th of 5 cps, which is validated by Annegarn et al. for chronic organ failure patients [22]. The sensor orientation threshold is mostly affected by the wear location. Like in the application presented by Annegarn et al., the sensor in the current application was attached to the upper leg with a plaster [22]. Therefore, the sensor orientation threshold did not have to be optimized for the application. 

Agreeing with Bersch at al., we expect other algorithm parameters to play a far less significant role in the classification process compared to the adjustable parameters described above [17]. Literature already established the effect of the sampling rate on the classification performance and concluded that a sampling rate of 25 Hz is sufficient to successfully classify ADLs [17]. To eliminate noise acceleration, a moving average with a window size of 0.12 s is used. This parameter is sensor and sample rate specific, making an optimization not relevant at this point. The settings of the frequency filters, although important, are beyond the scope of the current work.

The use of a simple decision tree classifier that has been successfully used in the past to discriminate between broad categories of physical activity e.g., dynamic, standing and sedentary behavior, is the first strength of the current work. According to Preece et al. the direct relation between the processing steps and thresholds in the decision tree on the one hand and the outcome measures on the other hand is one of the benefits of this classifier method [21]. Allahbakhshi et al. state that this makes it easier to understand and interpret compared to other classifier methods [34], a prerequisite for an adjustable algorithm. To discriminate more specific categories of physical activity e.g., cycling or stair walking, a machine learning approach might be more suitable [35,36]. The results of the current work confirm the need for SFP, including target population specific ADLs, in validation procedures in accordance with the recommendation of Lindemann et al. [11]. Looking at the classification error for FAP, one could conclude that there is no difference between the algorithm settings used by Annegarn et al. and the optimized algorithm settings in the current application [22]. However, for SFP the classification error for the algorithm settings used by Annegarn et al. for dynamic and standing behavior is too high and shows that, for the current application, these settings are inaccurate in a real-life situation [22]. Previously it has been shown that the target group accumulates the amount of physical activity primarily by household activities (50 minutes per day) [37]. The second strength of this paper is that the validation of SFP was carried out with the target population’s specific ADLs and, therefore, was representative of real life. Furthermore, the use of a gold standard to complement the good inter-observer reliability assures the quality of the suggested methodology.

There are some limitations in the current work that could be addressed in future research. First and most important, for SFP the error for measuring dynamic behavior was slightly higher than acceptable. This higher error partly originates from a poor rater behavior definition of the included ADLs. Setting the table, for example, is defined as a dynamic behavior, while in practice this is a sequence of dynamic and standing behaviors. Rather than defining an ADL as one type of behavior, the observers should classify the actual activity of the participant to possibly improve the classification accuracy [38]. Given this limitation, the classification error for dynamic behavior, therefore, presents the worst case scenario. Second, the amount of participants in each sub-group was limited. Ideally, validation studies should include sufficient participants with a range of physical activity behaviors. However, the overall sample size of forty participants in the current work is comparable to other validation studies [22,39,40,41]. Third, the three adjustable parameters were treated independently during the optimization procedure: the data segmentation window size was optimized for a fixed amount of physical activity threshold and, next, the amount of physical activity threshold was optimized for the optimal window size. This approach is suited for finding an optimal performance that meets the criteria of errors below 10%. However, to find the absolute optimal performance, all different window sizes could be tested for all amounts of physical activity thresholds, or other optimization operations could be applied. This could be interesting for future studies but does not add valuable insights for the aim of the current work.

Presently, parameter settings of the adjustable algorithm have been optimized and validated for one specific target population and one location for tracker wear. To prove the adjustable algorithm’s contribution to the fast realization of new applications, future work should focus on optimizing these parameters for other target populations and wear locations. New challenges are the validation for healthy young subjects and/or a chest wear location.

## 5. Conclusions

An adjustable physical activity classification algorithm that can discriminate between dynamic, standing, and sedentary behavior was transparently described.

The adjustable algorithm was successfully applied for a healthy elderly population and a tracker worn on an upper leg location under simulated free-living conditions.

By means of easily adjustable parameters, the algorithm performance can be optimized for different target populations and wear locations.

We propose that this method should be adopted by developers of activity trackers to facilitate the activity tracker selection process for researchers and clinicians. Furthermore, we are convinced that the adjustable algorithm could potentially contribute to the fast realization of new applications in terms target populations and/or wear locations.

## Figures and Tables

**Figure 1 sensors-19-05344-f001:**
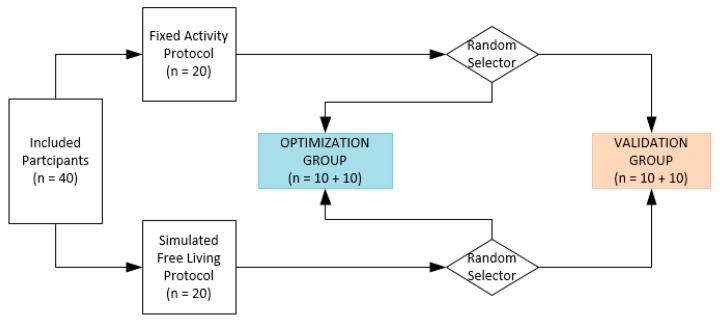
Two groups of twenty participants were random assigned to the optimization or validation group.

**Figure 2 sensors-19-05344-f002:**
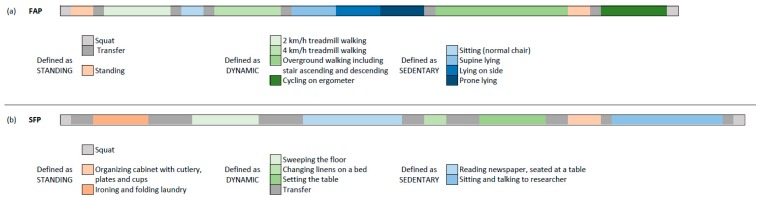
Graphical representation of (**a**) FAP with a predefined order and duration of the listed activities and (**b**) SFP where participants are free to choose the order and duration of the listed activities.

**Figure 3 sensors-19-05344-f003:**
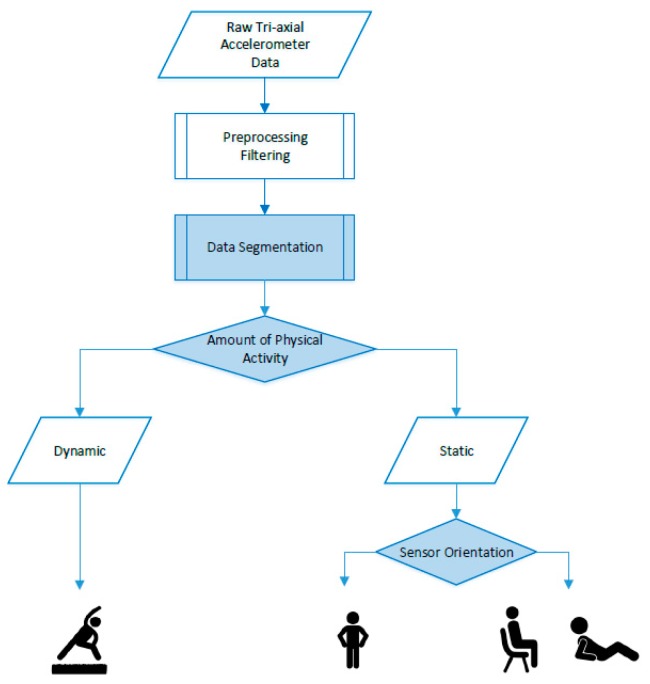
Schematic overview of the physical activity classification algorithm for the tracker worn on an upper leg location. The adjustable parameters are highlighted in blue.

**Figure 4 sensors-19-05344-f004:**
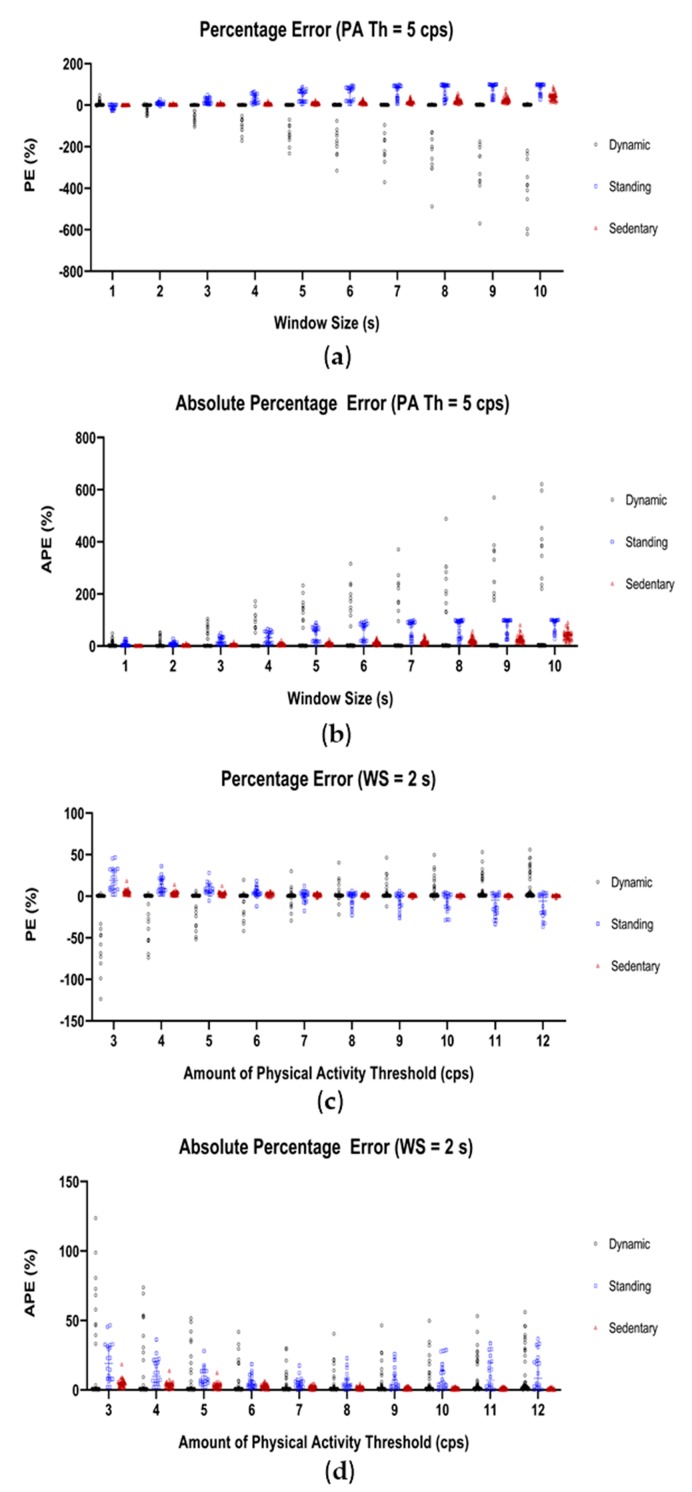
Results of the parameter setting optimization: (**a**) PE and (**b**) APE for the evaluated data segmentation window sizes, (**c**) PE and (**d**) APE for the amount of physical activity threshold. PE and APE for dynamic are presented in black, for standing in blue and for sedentary in brown.

**Figure 5 sensors-19-05344-f005:**
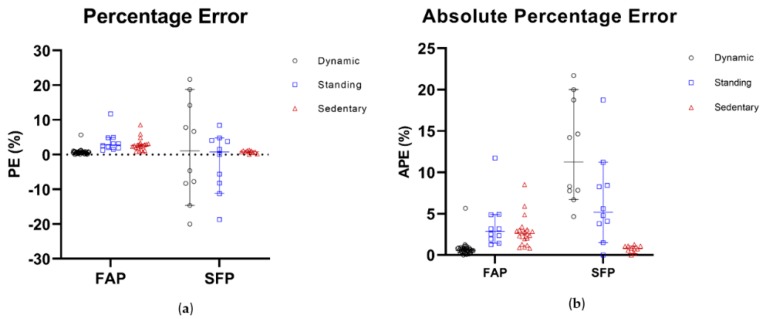
Percentage Error (**a**) and Absolute Percentage Error (**b**) for the validation data set. PE and APE for dynamic are presented in black, for standing in blue and for sedentary in brown.

**Table 1 sensors-19-05344-t001:** Participant characteristics for the optimization and the validation group.

**Fixed Activity Protocol**		
	**Optimization Group (n = 10)**	**Validation Group (n = 10)**
Gender (male)	5 (50%)	3 (30%)
Age (years)	69 (67–77)	68 (66–73)
Body weight (kg)	81.0 (63.0–90.0)	69.4 (59–86.5)
Body length (cm)	170 (159–175)	160 (157–169)
Average Walking Speed (m/s)	1.22 (0.88–1.60)	1.37 (1.06–1.48)
**Simulated Free-Living Protocol**		
	**Optimization Group (n = 10)**	**Validation Group (n = 10)**
Gender (male)	6 (60%)	4 (40%)
Age (years)	73 (70–77)	75 (69–88)
Body weight (kg)	88.1 (75.1–114.0)	70.0 (48.8–101.6)
Body length (cm)	174 (165–184)	169 (158–182)
Average Walking Speed (m/s)	1.16 (0.79–1.29)	1.15 (0.85–1.38)

**Table 2 sensors-19-05344-t002:** PE and APE results for the optimal window size and amount of physical activity threshold.

**Optimal Window Size**				**Optimal Amount of Physical Activity Threshold**			
**Percentage Error (%)**				**Percentage Error (%)**			
WS (s)	Dynamic	Standing	Sedentary	PA Th (cps)	Dynamic	Standing	Sedentary
**1**	1.5 (0.7–2.5)	−3 (−15.1–1.7)	1.3 (0.7–1.4)	**6**	0.7 (0.4–0.9)	3.5 (2.0–6.5)	2.4 (1.8–3.4)
**2**	0.6 (0.4–0.7)	6.8 (4.4–13.4)	2.9 (1.8–4.2)	**7**	0.7 (0.6–0.9)	1.9 (−2.3–4.0)	2.0 (0.5–2.5)
**3**	0.7 (0.4–0.9)	16.9 (8.2–31.8)	4.6 (2.9–5.8)	**8**	0.9 (0.6–1.0)	1.2 (−7.7–2.7)	1.4 (0.4–1.8)
**Absolute Percentage Error (%)**				**Absolute Percentage Error (%)**			
**WS (s)**	**Dynamic**	**Standing**	**Sedentary**	**PA Th (cps)**	**Dynamic**	**Standing**	**Sedentary**
**1**	1.6 (0.9–2.6)	4.7 (1.9–15.1)	1.4 (0.8–1.6)	**6**	0.9 (0.7–1.0)	4.0 (2.4–7.2)	2.4 (1.8–3.4)
**2**	0.9 (0.6–1.0)	6.8 (5.1–13.4)	2.9 (1.8–4.2)	**7**	0.9 (0.7–1.0)	3.9 (2.2–6.2)	2.0 (1.4–2.5)
**3**	1.0 (0.7–1.3)	16.9 (8.2–31.8)	4.6 (2.9–5.8)	**8**	0.9 (0.7–1.0)	3.6 (2.0–7.7)	1.7 (0.6–2.0)

**Table 3 sensors-19-05344-t003:** Validation results for FAP and SFP reported as PE and APE for the optimized algorithm settings and the algorithm settings used by Annegarn et al. [22].

**Optimized Algorithm Settings (WS = 2 s, PA Th = 7 cps, SO Th = 0.8 g)**						
	**Dynamic**		**Standing**		**Sedentary**	
	**FAP**	**SFP**	**FAP**	**SFP**	**FAP**	**SFP**
**PE (%)**	0.6 (0.4–0.8)	1.0 (−14.6–18.6)	2.9 (1.5–4.9)	0.8 (−11.2–4.8)	2.6 (2.0–3.0)	0.8 (0.2–1.1)
**APE (%)**	0.6 (0.4–0.8)	11.3 (6.7–20.0)	2.9 (1.5–4.9)	5.2 (1.5–11.2)	2.6 (2.0–3.1)	0.8 (0.2–1.1)
**Algorithm Settings Used by Annegarn et al. (WS = 1 s, PA Th = 5 cps, SO Th = 0.8 g)**						
	**Dynamic**		**Standing**		**Sedentary**	
	**FAP**	**SFP**	**FAP**	**SFP**	**FAP**	**SFP**
**PE (%)**	1.1 (0.6–2.9)	27.9 (11.6–40.0)	1.6 (1.5–2.4)	−12.6 (− 27.2–7.9)	1.7 (1.2–2.0)	0.3 (−0.1–1.0)
**APE (%)**	1.1 (0.6–2.9)	27.9 (11.7–40.0)	1.6 (1.5–2.4)	12.6 (7.9–27.2)	1.7 (1.2–2.0)	0.4 (0.1–1.0)

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
