# Peer review of "Optimization and Validation of an Adjustable Activity Classification Algorithm for Assessment of Physical Behavior in Elderly"

_sensors, 2019, doi:10.3390/s19245344_

Round 1

Reviewer 1 Report

1. The overall structure of the paper can be adjusted:

(1) The introduction at the beginning of the paper is too much and can be modified as appropriate. Or you can refine some of the content and put it into the related work;

(2) The fourth part of the article discusses too much content, and the relevant content should be succinctly described in the conclusion section.

2. The idea of the article is interesting, but the algorithm is too simple. Every part of the algorithm is very basic, so that no other innovations and good results are seen.

3. The article lacks the experimental comparison of this algorithm with other similar algorithms. The results of this algorithm should be compared to the results of other similar algorithms to prove that your algorithm is better.

Reviewer 2 Report

I was honored to review the manuscript entitled "Optimization and validation of an adjustable activity classification algorithm for assessment of physical behavior in elderly" submitted to Sensors.

Taking into account the multiple studies ongoing in this field this type of study is needed.  I have only few small remarks that authors should adress properly.

I recommend to accept the manuscript after minor revision.

Points that need correction:

 - please provide the list of abbreviations.

 - please clearly provide the aim of this study

 - Introduction and Discussion section needs improvement- please cite doi: 10.1097/MD.0000000000014909. ; Biomed Environ Sci. 2016 Oct;29(10):706-712. ; 10.1016/j.biopha.2016.02.017.

 -  in discussion please provide the study strong points and limitations

I recommend to accept the manuscript after minor revision.

Reviewer 3 Report

This is a well-presented paper with a sound rationale. It is good to see the signposting to the reader (especially students) what activity trackers are actually measuring, and an appropriate insight in activity trackers. The purpose and justification of the study is clear. The paper appears to be a good methodology paper as well. Perhaps the authors could signpost this as well, in the introduction or discussion.

(1) A bit more background would be helpful to understand the weakness with current physical activity classification algorithm that discriminating dynamic, standing and sedentary behaviour.

(2) It would be appropriate if the authors could formulate their working hypothesis(ses), clearly linked with the research aims.

(3) Could the authors clarify what they consider as sedentary behaviour and how was this assessed, if this is necessary at all.

(4) Could the authors provide more details regarding the procedure for inter-rater reliability.

(5) The discussion looks rather descriptive. The research problem could be re-evaluated in light of the results obtained, and a more critical discussion is warranted.

Round 2

Reviewer 1 Report

I am honored that my opinion can help you improve your paper. The improved paper has been greatly improved compared to the previous one, but there are still some problems:

1.In the content of your changes, there are still some language structure problems, please correct them carefully.

2.If you can, design more contrastive experiments to prove the superiority of your algorithm.

3.I still recommend that you include the "related work" section in the paper to introduce some of the work in this area, which can provide readers with a relevant knowledge base. You can refer to the following articles, which may be helpful to you, and you can refer to them in the relevant work in the paper.

   1>.Liu L , Wang S , Hu B , et al. Learning Structures of Interval-Based Bayesian networks in Probabilistic Generative Model for Human Complex Activity Recognition[J]. Pattern Recognition, 2018, 81.

   2>.John A. Batsis, John A. Naslund, Lydia E. Gill, et al. Use of a Wearable Activity Device in Rural Older Obese Adults: A Pilot Study[J]. Gerontology & Geriatric Medicine, 2016, 2.

   3>.Baig M M , Gholamhosseini H , Connolly M J . A comprehensive survey of wearable and wireless ECG monitoring systems for older adults[J]. Medical & Biological Engineering & Computing, 2013, 51(5):485-495.
